# How Acculturation Influences Attitudes about Advance Care Planning and End-of-Life Care among Chinese Living in Taiwan, Hong Kong, Singapore, and Australia

**DOI:** 10.3390/healthcare9111477

**Published:** 2021-10-30

**Authors:** Fu-Ming Chiang, Ying-Wei Wang, Jyh-Gang Hsieh

**Affiliations:** 1Department of Nursing, Hualien Tzu Chi Hospital, Buddhist Tzu Chi Medical Foundation, Hualien 97002, Taiwan; mingfly6786@yahoo.com.tw; 2Department of Family Medicine, Hualien Tzu Chi Hospital, Buddhist Tzu Chi Medical Foundation, Hualien 97002, Taiwan; drywwang@gmail.com; 3Department of Medical Humanities, School of Medicine, Tzu Chi University, Hualien 97002, Taiwan

**Keywords:** advance care planning, culture, cross culture research

## Abstract

Background: Understanding attitudes towards life and death issues in different cultures is critical in end-of-life care and the uptake of advance care planning (ACP) in different countries. However, existing research suffers from a lack of cross-cultural comparisons among countries. By conducting this comparative study, we hope to achieve a clear understanding of the linkages and differences among healthcare cultures in different Chinese societies, which may serve as a reference for promoting ACP by considering cultural differences. Methods: Our researchers recruited Chinese adults who could communicate in Mandarin and lived in metropolitan areas in Taiwan, Hong Kong, Singapore, and Australia. Focus group interviews were conducted, and the interview contents were recorded and subjected to thematic analysis. Results: Between June and July 2017, 14 focus groups with 111 participants were conducted in four regions. With traditional Chinese attitudes towards death as a taboo, many participants felt it would be challenging to discuss ACP with elderly family members. Most participants also desire to avoid suffering for the self and family members. Although the four regions’ participants shared a similar Chinese cultural context, significant regional differences were found in the occasions at which participants would engage in end-of-life discussions and select settings for end-of-life care. By contrast, participants from Singapore and Australia exhibited more open attitudes. Most participants from Taiwan and Hong Kong showed a preference for end-of-life care at a hospital. Conclusions: The developmental experiences of ACP in Western countries, which place a strong emphasis on individual autonomy, cannot be directly applied to family-centric Asian ones. Healthcare professionals in Asian societies should make continuous efforts to communicate patient status to patients and their family members to ensure family involvement in decision-making processes.

## 1. Introduction

The concepts of end-of-life care and advance care planning (ACP) originated from Europe and the United States, with the majority of related research driving from Western countries. As vast differences exist between the concept of family and individual autonomy between Asian and Western cultures, the developmental experiences of Western countries cannot be directly applied. For example, in the medical decision-making process for Asian parents, the decision of the family usually takes precedence over an individual’s decision [1]. The ACP process involves medical decision-making and end-of-life discussions between the health provider and the individual. However, this process conflicts with the tradition of family involvement in medical decision-making among Asians, leading to difficulties in the uptake of ACP [2,3]. Moreover, in traditional Chinese culture, a conversation about a person’s own death and dying would not be initiated until the person faced a terminal illness. Open discussions about death are considered a cultural taboo and regarded as a bad omen [4]. Even among the geographically proximate countries of South Korea, Japan, and Taiwan, where Asian cultures are dominant, significant differences exist in attitudes towards end-of-life care due to differences in traditional cultures, values, religion, and health care systems [5]. Acculturation is a process through which a person or group from one culture comes to adopt the practices and values of another culture, while still retaining their own distinct culture [6]. Culture is often described as a laundry list of phenomena: norms, beliefs, worldviews, values, rituals and practices. Importantly, culture is not merely an overlay on existing social relations, it is constitutive of social relations and is a structuring force that shapes interactions at and across all levels of social life [7]. People with the same cultural background may change their views regarding end-of-life care after migrating to other countries due to the influence of factors such as social values and language barriers [8,9]. This study conducted focus groups in Japan with English-speaking Japanese Americans, Japanese-speaking Japanese Americans, and local Japanese citizens to compare end-of-life care and palliative care services [10]. The preferred decision-making models of all the groups involved family members as participants or main decision-makers, but English-speaking Japanese Americans regarded individual participation and opinion expression as a critical component. Similarly, the study reported that English-speaking Japanese Americans had more positive attitudes towards foregoing treatment at the end-of life, participating in ACP, and autonomous decision-making practices [11]. Understanding of attitudes towards life and death issues in different cultures is critical in end-of-life care. Lin analysed the pursuit of the meaning of end of life among Chinese immigrants in the U.S. and concluded that the understanding of patients’ values towards life and death is greatly beneficial to the provision of appropriate cross-cultural nursing care [12]. This is consistent with the findings of a study by Sinclair, et al. [13] who reported respecting and understanding cultural backgrounds, and the provision of appropriate information for decision-making constitute the first step in discussing end-of-life issues among migrant groups. An effective ACP program includes repeated interactive discussion sessions, decision aids, and interventions targeting multiple stakeholders, which are closely associated with the local socio-cultural environment and health system [14]. Many studies have shown that cultural barriers pose as the greatest challenge towards the uptake of ACP in Asian countries [15,16,17]. The existing research suffers from a lack of cross-cultural comparisons among countries, leading to difficulties in understanding cultural contexts and their subtle influences on adopting ACP [18,19]. By conducting this comparative study, we hope to achieve a clear understanding of the attitude towards ACP in different Chinese societies, which may serve as a reference for the promotion of ACP in other countries.

## 2. Materials and Methods

The study aims to understand the impact of cultural integration of Chinese ethnic groups with different levels of westernization on the promotion of advance medical plans, and the key factors in promoting ACP: When is the right time? Where is the right place? Who is the right person to discuss? What is the right content? How to proceed? The focus group method is a useful and effective mechanism for deriving collective opinions, values and beliefs. This qualitative method offers the advantage of allowing researchers to obtain in-depth comments and feedback from participants in a more proactive, semistructured and interactive manner [20]. Transnational focus group interviews in the four regions with different proportions of Chinese (Taiwan: 99%, Singapore: 76.2%, Hong Kong: 92%, and Australia: 5.6%) were conducted by three moderators independently. The focus group process was discussed through online meeting to achieve consensus and consistency. The inclusion criteria for participants in this study were as follows. (1) Chinese adults over 50 years old who could communicate in Mandarin/Cantonese and were the first generation of immigrants (over 30 years of residence). (2) Chinese adults who could communicate in Mandarin/Cantonese, over 40 years old and who received formal education of elementary school and above in the place of study. (3) Chinese adults who could communicate in Mandarin/Cantonese, over 40 years old, and the second generation of immigrants. All participants were volunteers and informed consent was received. Each focus group included 8–10 participants and lasted for 1.5–2 h. The study took place in the local community in different study regions, i.e., in the community center.

### Data Collection and Analysis

All sessions were audiotaped and then transcribed verbatim in the language of the group and subjected to thematic analysis. All participant data were delinked to ensure anonymity. The transcripts were read multiple times and meaningful content was marked for inductive analysis and identification of key themes by two researchers. After comparison of the results, differences were discussed, and a consensus was reached. During the second round of analysis, a check was performed to ensure that all statements fitted the categories and statements were recognized. The study is reported in accordance with the COnsolidated criteria for REporting Qualitative research (COREQ) [21]. 

## 3. Finding and Results

Between June and July 2017, 14 focus groups with a total of 111 participants were conducted in Taiwan, Singapore, Hong Kong, and Australia (Appendix A). The findings from the focus groups are summarized (Appendix A). No participants dropped out during the study period.

### 3.1. People with Whom the Participants Wished to Discuss about ACP—“Who”

In general, the participants from the various regions wished to discuss issues regarding their acceptance of end-of-life care with their family members. Most participants would engage in discussions with their children. A small number indicated that they would discuss it with their spouses, but almost none of the participants mentioned their intention to discuss it with their elderly family members. Participants from Taiwan frequently mentioned the need to obtain the opinions of doctors or healthcare professionals before discussing it with their family members, which demonstrates the difference in the degree of respect towards and reliance on healthcare professionals between Taiwan and other regions.

‘Nowadays, I tell them [my children] about my healthcare wishes for the future, I think that radiotherapy and chemotherapy are a waste of resources and will impose a mental burden on them. I hope they will respect my wishes.’ (Participant S4 from Singapore)


*‘The doctor is professional, so he knows how severe your condition is and what kinds of treatment you will need. I will discuss with the doctor; I think I need to consult a professional before compromising with my children.’ (Participant TC2 from Taiwan)*



*With traditional Chinese attitudes towards death such as ‘death is a taboo’ and ‘death is ominous’, many participants felt that it would be difficult to discuss ACP with elderly family members. The participants tended to rely on the traditional Chinese concept of ‘men are superior to women’. Therefore, certain participants expressed the view that they would ask a male relative, which is usually the eldest son, to assume the responsibility.*



*‘I said, “You should tell our elder brother and not just tell me, because this should be known to everyone, not just me.’ Before my father passed away, I was the one taking care of him, and he had already told me everything regarding his end-of-life matters. My mum and elder brother are still around, I have an elder sister too…and a younger brother. You know, men are considered superior in Chinese culture, I said that I have no right to express my opinion.’ (Participant S1 from Singapore)*


### 3.2. Occasions on Which Participants Would Discuss End-of-Life Care with Their Family Members—“When”

Although the participants from the four regions shared a similar Chinese cultural context, significant regional differences were found in the occasions at which participants would engage in end-of-life discussions. Participants in Taiwan were inclined towards specific occasions to initiate such discussions. These occasions included the following: becoming aware that someone they know had a serious health condition, coming across a discussion of a related topic on a television program, or after seeing a cemetery advertisement during a trip. Participants in Hong Kong expressed similar views about the necessity of specific occasions. It was mentioned that people usually made use of family gatherings to initiate discussions with their children through indirect methods, such as reflection or hinting, when the care situations of other relatives were being discussed.


*‘In Hong Kong, it is inevitable that multiple generations live together, so it is more appropriate [to discuss] at family gatherings or when everybody comes back for a meal. There are not many occasions to talk about such matters out of the blue. Most of the time, the conversation is about having a patient at home, or someone saw a great-uncle in a bad condition [referring to the end-of-life care situation] …these are occasions to initiate discussion.’ (Participant HK2 from Hong Kong)*


By contrast, participants from Singapore and Australia exhibited significantly more open attitudes. They were willing to engage in end-of-life discussions on any occasion, e.g., during hospitalisation or everyday life at home.


*‘I think it is more appropriate to discuss with your family during everyday life, because you won’t know when you will encounter such a situation. Sometimes when you sit down and chat with your family members, you can discuss such issues. Nowadays, people have fewer children, so it will not be troublesome. With direct communication, all family members will have the same understanding.’ (Participant S2 from Singapore)*


In fact, once the discussion about ACP had been initiated, the reactions of the participants’ family members did not differ among the various regions. Additionally, the majority of elderly or junior family members involved in the discussion did not view the topic as taboo, which was contrary to expectations.


*‘On one occasion, I was having tea with my mum and aunt, and I started to talk about it. The mood during the discussion was great and I realised that all three elderly family members had their own wishes. However, the mood should not be too sombre during the discussion.*
*’ (Participant HK3 from Hong Kong)*


### 3.3. Desired Medical Treatments towards the End of Life and the Concept of ACP—“What”

The participants from the various regions were inclined towards receiving palliative and supportive care for the reduction of pain and increase of physical comfort towards the end of life. They were not receptive to life-sustaining treatments that prolonged the near-death process, including intubation, defibrillation, and cardiac massage, as they felt that such treatments not only increase the pain of the patient but also cause distress for their relatives.


*‘I told my husband that I do not wish to be resuscitated when life reaches the stage where I become unaware and unconscious, as it is meaningless and prolongs suffering; it puts a burden on my family and makes them suffer too.’ (Participant HK5 from Hong Kong)*


The participants from Singapore had a clearer understanding and positive attitude toward of ACP than the other regions. Participants could clearly point out the various items and concepts about ACP, and were willing to sign ACP. Polices support by government and creation of a positive social atmosphere were helpful to promote ACP.


*‘The government promotes the concept of ACP through TV station and social networks, and especially in senior community center. When the elderly has a clear understanding of the concept of ACP, they will accept it and will gradually implement it step by step. With this kind of preparation, I feel very lucky because the government has been promoting these things.” (Participant SG2 from Singapore)*


### 3.4. Preference of Care Setting at the End-of-Life—“Where”

Differences in opinion regarding the setting for end-of-life care existed among the participants from different regions. Participants from Singapore had divided opinions, with some showing a preference for end-of-life care at home due to the visitor restrictions at hospitals and the inconvenience of travelling back and forth from the hospital for their family members. Other participants preferred to pass away in a hospital or hospice care unit in view of the lack of healthcare professionals and medical equipment at home, the complicated process of filing out a death certificate for a home death, and negative emotions such as fear and a sense of loss that may arise in young children.


*‘Only four visitors are allowed in the hospital ward, so if the patient has visitors, she [the nurse] will tell you that there are visitors waiting downstairs. Those visitors will have to wait for their turn to visit, which is very inconvenient. There are no visitor restrictions if the patient is at home, and the visitors can be at ease when they visit.’ (Participant S5 from Singapore)*


Most participants from Taiwan and Hong Kong showed a preference for end-of-life care at a hospital, with the reasons differing between the regions. Participants from Hong Kong stated that although public hospitals offered the advantage of lower cost compared to private hospitals, the length of the hospital stay was also restricted. In addition, family members would have to bear the additional cost of bed reservation at a nursing home during the hospitalisation period. Some participants preferred at-home care by a domestic helper if circumstances permitted. However, they would opt to seek medical treatment and pass away in a hospital setting in case of an emergency, as the process for filing out a death certificate for a home death is relatively complicated. Participants from Taiwan preferred end-of-life care by a palliative care team at a hospital as they trusted hospice care personnel and did not wish to increase the burden on their children.


*‘Hospitals will provide the evidence of death so that you can obtain the certificate of death. If someone passes away at home, there are more troublesome matters to deal with. When my uncle died, the certificate of death was only issued one month later, his family members had to go to court and seek a lawyer, it’s very troublesome…’ (Participant HK3 from Hong Kong)*


## 4. Discussion

The aim of this study was to achieve a clear understanding the preference of end-of-life care in different Chinese societies, which may serve as a reference for promoting ACP by considering cultural influences. The study confirms Chinese societies place great emphasis on group and family decision-making [21,22,23]. In situations where decision-making is required for end-of-life care, personal autonomy and self-determination do not take precedence; instead, there is a strong reliance on doctors and family members, and authoritative healthcare experts are usually entrusted with the final say. In this study, participants from Taiwan expressed the belief that the professional opinions of doctors had to be sought before discussing ACP with their family members. Participants from other regions also indicated the need to discuss it with their children and spouses before making a decision, which is consistent with the results of previous studies that investigated the differences in end-of-life care decision-making processes between U.S. and Asian populations [24,25]. In a study by Chu, et al. [26] on 2878 hospitalised patients eligible for ACP, it was found that the participation of healthcare professionals in ACP discussions was a key factor for the completion of advance directives in the subject, and the participation of family members was a key factor for end-of-life care decision-making. These findings are highly consistent with the results of the present study.

The concepts of high-context cultures (HCs) and low-context cultures (LCs), which were introduced by the anthropologist Edward T. Hall in 1959, can be used for the systematic classification of cultures [27]. Within HCs, the interpretation of messages relies on cues arising from cause-effect relationships. Members rely heavily on nonverbal cues (body language, silence, facial expressions, etc.) when communicating with others, utilise indirect and roundabout methods in their thinking processes and conveyance of information, and interact with each other under the influence of their relationships. By contrast, members of LCs utilise clear, direct spoken and written words for communicating and conveying messages and tend to emphasise logic and argumentation [28]. The cultural contexts of countries are not absolutely high or low but relative to those of other nations. For instance, countries in the East generally have HCs compared with Western countries. Hall’s contexting model for intercultural communication has been used widely; however, there are some critiques about the development of this model [29,30]. In the present study, participants from Hong Kong and Taiwan were inclined towards the utilisation of specific occasions and scenarios for the initiation of end-of-life care discussions, whereas participants from Singapore and Australia did not exhibit such an inclination. This is consistent with the findings of Chen and Wang [31], who found that compared with countries with a Western culture, countries with an Eastern culture or a blend of Eastern and Western cultures were inclined towards the communication styles of HCs, which involve the utilisation of contexts, cause-effect relationships, and other metaphorical methods to interpret conveyed messages. By contrast, countries with a Western culture were inclined towards the communication styles of LCs, which involve the direct communication of arguments. Among the four regions investigated in this study, Taiwan and Hong Kong have predominantly Chinese societies. Participants from these two regions indicated that they would initiate ACP discussions by utilising similar situations faced by others or appropriate opportunities (such as changes in circumstances) to subtly express their wishes regarding end-of-life care, which is consistent with the indirect communication methods of the HCs in Eastern countries. In Singapore and Australia, which have been influenced by Western cultures to a greater extent, the participants indicated that they would utilise direct conversations to discuss their wishes regarding end-of-life care with their family members, which is in line with the direct communication styles of the LCs in Western countries. Therefore, our findings provide a clear explanation of the differences in communication contexts among different regions [27,28].

In recent years, high-quality end-of-life care has received increasing attention due to the issue of global aging. Participants from various regions had different considerations when selecting the desired setting for end-of-life care, which included family financial status, social benefits and public health insurance, the need for professional healthcare, the feelings of family members, and the convenience of family visitation. Gomes and Higginson (2006) [32] reported that medium or high social class, and living with relatives, were key influencing factors of the selection of home care in terminally ill patients with cancer. In addition, communication and coordination among family members [33] and the degree of complexity of the caregiving [34] also influenced the choice of setting for end-of-life care.

## 5. Conclusions

The degree of acculturation plays an important role when discussion ACP among Chinese living in different region. The preference for time, place, content, and the way for discussing ACP varies in the four study regions. However, parents in Chinese societies often provide unconditional support for their children and hope to ease their financial and emotional burdens. When selecting end-of-life program, the desire to avoid inconvenience to their family members took precedence over their own wishes. Healthcare professionals in family-values societies should make continuous efforts to ensure family involvement in decision-making processes.

## Data Availability

The data presented in this study are available on request from the corresponding author. The data are not publicly available due to privacy.

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
