# Peer review of "How Acculturation Influences Attitudes about Advance Care Planning and End-of-Life Care among Chinese Living in Taiwan, Hong Kong, Singapore, and Australia"

_healthcare, 2021, doi:10.3390/healthcare9111477_

Round 1
Reviewer 1 Report
This is an interesting study on understanding attitudes towards end-of-life care and the uptake of advance care planning (ACP) amongst ethnic Chinese people in four regions. I believe that the focus of this study makes an important contribution to both the intercultural and healthcare literature, and will be great interest to the readership of the journal. That said, there are a number of ways in which the manuscript could be improved further. I outline these below with examples where appropriate.
Introduction:
Although the introduction sets the scene nicely, it is unclear what perspective on ‘culture’ the authors are taking. The notion of ‘culture’ is contested in the cross- and intercultural communication literature, and a more critical approach would be desirable. For example, the notions of ‘East’ and ‘West’ need to be unpacked and approached from a more critical perspective. The authors make several generalisation about ‘traditional Eastern cultures’ and ‘Chinese societies’ but it is not clear who/what exactly is meant by this. It would be useful to include a definition of ‘culture’ here too. Conceptual clarity needs to be improved and key terms need defining e.g. ‘acculturation’ is used in the title of the manuscript but not explained in-text. The authors refer to Chinese attitudes (e.g. towards death) multiple times, but some links to the literature could be added here to explain this further.
Methods:
Some proofreading is needed in this section as there are some apparent language issues and typos (e.g. ‘methods and methods’). More clarity is needed on how exactly and through which channels participants were recruited. What kinds of questions/prompts were used in the focus groups? More focus groups were held in Taiwan than other regions – how might this have affected the findings? What was the relationship between the focus groups and the semi-structured interviews, and what was the rationale for using both? A clearer rationale could also be provided as to why the four regions were selected. Some background information would be useful (perhaps best placed in the introduction). Were the data transcribed verbatim? Were data transcribed in Mandarin/Cantonese and selected excerpts subsequently translated? How did the authors go about this? What relationship did the interviewer/moderator have to the participants? In which locations were the focus groups/interviews conducted? Finally, the authors mention that they used an inductive approach in their thematic analysis. Did they use a particular framework (e.g. Braun and Clarke, 2006), and how exactly were key themes derived from the data?
Results:
There are some formatting issues here (font). The ‘who’, ‘when’, ‘what’, ‘where’ structure works well for data presentation – was this the analytical framework for the data analysis? The authors claim that participants shared a ‘similar Chinese cultural context’ (p. 3, 4) – how and to what extent is this true when they lived in different regions? This could be explicated further. The ‘what’ section is very short relative to the other sections.
Discussion and conclusions:
The authors make some fairly crude generalisations here, and some claims need to be substantiated with links to the literature. Examples include:
‘Chinese societies place great emphasis on group and family decision-making.’ (p. 5), and notions of ‘East’ and ‘West’ and HC/LC cultures.
There is a substantial amount of literature that critiques these notions – this should at least be acknowledged here.
In the conclusion section, the authors could be more explicit about implications for healthcare.
Author Response
Dear Reviewer,
Good Day!
According to your comments, we have revised our manuscript ‘How acculturation influences attitudes about advance care planning and end-of-life care among Chinese living in Taiwan, Hong Kong, Singapore, and Australia’, and now submitting again to healthcare to be considered for publication.The detailed responses are as follows.

Reviewer 2 Report
‘Methods and methods’ – Review this heading
This section seems to be have missed when being checked for language – there are a number of issues, this section will need to be edited eg.
Second sentence – ‘Focus group interviews in the four regions were conducted by three moderator independently’ - should it be ‘…three independent moderators’? Or ‘…conducted independently by three moderators’?
Forth sentence – ‘Participants recruitment and interview by three moderator independently’
Methods - This is a good study design and appears to be appropriate and well carried out but this section needs further explanation to really know this – why focus groups? (there is no comment on methodology and why focus groups were the most appropriate method for the study.) How many were involved in focus groups and how many in interviews? Or were these the same people, taking part in a focus group and then an interview? Why were there both? Was this a pragmatic issue because not everyone could attend the focus groups? How many focus groups in each region? The reader need to get a better feel for why the study was designed in this way and how it was carried out.
Analysis – There will have been a large amount of data generated by these focus groups so I think some more detail about how this was handled, who was involved in this and what their roles were would be of benefit
Findings - This is a qualitative study so the heading should be Findings, rather than Results
Some issues in this section with italicisation of quotes and explanatory text
Discussion – Perhaps the authors could take a more critical stance in their discussion – while they recognise that ACP cannot be simply transposed to different cultures they could also consider whether ACP is appropriate at all and instead consider what might constitute good end of life care in the Chinese context.
Conclusion - doesn’t seem to sum the discussion but rather add a new dimension to it. Feel this should be moved to the end of the discussion and a concluding paragraph surmising the key points be added.
Author Response

(The authors gave the same response as above.)

Reviewer 3 Report
I think just a brief review of how generalizable the authors believe their findings to be, and on what basis that claim is made statistically.
Otherwise, fascinating, very useful paper.
Author Response

(The authors gave the same response as above.)
